# An Exploration of Dynamic Changes in the Mulberry Growth Process Based on UPLC-Q-Orbitrap-MS, HS-SPME-GC-MS, and HS-GC-IMS

**DOI:** 10.3390/foods12183335

**Published:** 2023-09-06

**Authors:** Shufang Wu, Jiaxin Yin, Xuejuan Li, Jingyi Xie, Hui Ding, Lifeng Han, Songtao Bie, Fangyi Li, Beibei Zhu, Liping Kang, Xinbo Song, Heshui Yu, Zheng Li

**Affiliations:** 1College of Pharmaceutical Engineering of Traditional Chinese Medicine, Tianjin University of Traditional Chinese Medicine, Tianjin 301617, China; wsf199807@163.com (S.W.); yiyi526117@163.com (J.Y.); lixuejuan20221021@163.com (X.L.); xiejingyi199709@163.com (J.X.); dinghui.hn@163.com (H.D.); song9209@tjutcm.edu.cn (S.B.); lifangyi@tjutcm.edu.cn (F.L.); zhubb@tjutcm.edu.cn (B.Z.); songxinbo@tjutcm.edu.cn (X.S.); lizheng@tjutcm.edu.cn (Z.L.); 2Haihe Laboratory of Modern Chinese Medicine, Tianjin University of Traditional Chinese Medicine, Tianjin 301617, China; hanlifeng_1@sohu.com; 3State Key Laboratory of Component-Based Chinese Medicine, Tianjin University of Traditional Chinese Medicine, Tianjin 301617, China; 4State Key Laboratory Breeding Base of Dao-di Herbs, National Resource Center for Chinese Materia Medica, China Academy of Chinese Medical Sciences, Beijing 100700, China; kang_liping21@163.com

**Keywords:** mulberry, volatile organic compounds, non-volatile organic compounds, dynamic process analysis, UPLC-Q-Orbitrap-MS, HS-SPME-GC-MS, HS-GC-IMS

## Abstract

This work was designed to investigate the dynamic changes process of non-volatile organic compounds (n-VOCs) and volatile organic compounds (VOCs) in mulberries during different growth periods using UPLC-Q-Orbitrap-MS, HS-SPME-GC-MS, and HS-GC-IMS. A total of 166 compounds were identified, including 68 n-VOCs and 98 VOCs. Furthermore, principal component analysis (PCA), random forest analysis (RFA) and orthogonal partial least squares discriminant analysis (OPLS-DA) were used to analyze differences in mulberries at different ripening stages. A total of 74 compounds appeared or disappeared at different ripening periods and 24 compounds were presented throughout the growth process. Quantitative analysis and antioxidant experiments revealed that as the mulberries continued to mature, flavonoids and phenolic acids continued to increase, and the best antioxidant activity occurred from stage IV. Conclusively, an effective strategy was established for analyzing the composition change process during different growth periods, which could assist in achieving dynamic change process analysis and quality control.

## 1. Introduction

The mulberry plant (genus *Morus*, family *Moraceae*) is a perennial woody plant native to eastern and central China [1]. It is now widely cultivated in countries of East Asia such as China, Japan, and South Korea, and in south and north Africa. The mulberry is more than 7000 years old in China and is the most diverse species of mulberry in the world [2]. The mulberry, as a traditional natural plant, shows homology in use across medicine and food and holds great promise in both medicine and food development. On the one hand, mulberry plants have been used for thousands of years as a traditional Chinese medicine to treat diseases. The chemical constituents of mulberry fruits mainly include flavonoids, anthocyanins, phenolic acids, and polysaccharides [3]. Modern pharmacological studies had shown that it has hepatoprotective [4,5], antioxidant [6], anti-inflammatory [7], anticancer [8], hypotensive [9], neuroprotective [10], and immunomodulatory effects. Some compounds had been reported to be effective in modulating metabolic activity, such as moranoline, albafuran, albanol, morusin, kuwanol, calystegine, and hydroxymoricin [11,12]. On the other hand, mulberries also contain a variety of nutritional elements that play a crucial role in human metabolism. Mulberry fruits have a higher total amino acid ratio (TAA) compared to high-quality protein foods such as milk and fish [13]. Gundogdu et al. [14] reported that the glucose content in mulberry is about eight times that of sucrose, while the fructose content is about five times that of sugarcane. The protein content of fresh mulberries is higher than that of raspberries and strawberries, and the anthocyanin content is higher than that of blackberries, blueberries, blackcurrants, and red currants [15]. Mulberries are consumed in fresh or processed form due to their nutritional value, such as juice, fermented fruit wine, tea, yogurt, dried or natural pigments [16,17]. The composition of the mulberry plant has also gained increasing attention because of its rich nutritional value and biological activity.

In a previous study, ultra-performance liquid chromatography (UPLC) combined with a linear trap quadrupole and Orbitrap mass analyzer, diode array detector, and a triple-quadrupole mass spectrometer were used to analyze the composition of *Morus alba*, respectively [18]. Forty-one compounds were identified, including 14 hydroxycinnamic acid esters, 13 flavonol glycosides, and 14 anthocyanins. Thirty-one phenolic compounds from white and black mulberry leaves were identified by UHPLC-MS [19]. Phenolic constituents were characterized by the presence of a large number of flavonol derivatives, mainly in the glycosylated forms of quercetin and kaempferol. VOCs in mulberry fruits have also been reported. Free volatile compounds (VOCs) and glycoside-binding volatiles (GBVs) of four mulberry cultivars were investigated by solid-phase extraction (SPE) and HS-SPME-GC-MS [20]. Fifty-five VOCs and 57 GBVs were identified and quantified. Drying methods were found have effects on the flavor of mulberry fruits, as revealed by GC-MS and descriptive sensory analysis [21]. It is noteworthy that the heat-treated mulberry samples had sour aromatics, while the non-heat-treated samples had sweet aromatics.

To sum up, most of the reports selected only one of both of n-VOCs and VOCs for characterization. HS-GC-IMS is a powerful technique for the separation and sensitive detection of VOCs, which is characterized by rapid response, high sensitivity, simple operation, and low cost [22]. It combines the high separation capability of GC with the rapid response of ion mobility spectroscopy and is gaining popularity in drug detection, disease surveillance, and environmental protection, especially for food flavor analysis [23]. HS-GC-IMS tends to be used to identify samples and can also detect small odor molecules not detected by GC-MS. To the best of our knowledge, HS-GC-IMS has not been used to capture the flavor compounds of mulberry fruits. The aim of this study was to comprehensively characterize the n-VOCs and VOCs of mulberry fruits at different ripening stages and monitor the dynamic change process. The combination of UPLC-Q-Orbitrap-MS, HS-SPME-GC-MS and HS-GC-IMS provided a comprehensive analysis of n-VOCs and VOCs, and a complete analytical strategy was established. Exploring the changes of compounds during the growth of mulberry fruits provides basic data for the quality evaluation of mulberry fruits, and also provides references for the cultivation of high-quality mulberry fruits as well as the in-depth development and utilization of mulberry fruits. All the works of this study are presented in Figure 1.

## 2. Materials and Methods

### 2.1. Chemicals and Reagents

HPLC-grade acetonitrile and methanol were obtained from Fisher (Fair Lawn, NJ, USA), and formic acid was obtained from ACS (Wilmington, DE, USA). Experimental water was purchased from Watsons Trademark Co., Ltd. (Hong Kong, China). The reference standards of protocatechuic acid (CAS: 99-50-3), caffeic acid (CAS: 331-39-5), gallic acid (CAS: 149-91-7), quercetin (CAS: 117-39-5), taxifolin (CAS: 480-18-2), chlorogenic acid (CAS: 27-97-9), astragaloside (CAS: 480-10-4), isoquercetin (CAS: 21637-25-2), rutin (CAS: 153-18-4) and sangone G (CAS: 75629-19-5) were provided by Shanghai Yuanye Bio-technology Co., Ltd. (Shanghai, China). The standard mixture N-ketone C4-C9 for calculating the retention index (RI) of HS-GC-IMS was purchased from Sinopharm Chemical Co., Ltd. (Shanghai, China), and the normal alkane C8-C20 of GC-MS was purchased from Sigma Aldrich Chemical Co., Ltd. (St. Louis, MO, USA).

### 2.2. Sample Collection

Mulberries were obtained from the arboretum of Tianjin University of Traditional Chinese Medicine, Tianjin, China (116.97 °E, 38.95 °N) and were identified by Professor Lijuan Zhang (School of Chinese Materia Medica, Tianjin University of Traditional Chinese Medicine, Tianjin, China) as the fruits of *Morus* species of the *Moraceae* family *Morus*. Based on the shape of the mulberry tree and the appearance of its fruit, the variety of mulberry was roughly determined to be ‘Black Pearl’. Samples were collected in mid-April (stage I), early May (stage II), mid-May (stage III), and late May (stage IV) of 2022, respectively. Between 5 and 7 mulberries were selected randomly and marked to ensure that they were collected in each period. The collected samples were transported back to the laboratory in ice boxes, a part was immediately analyzed and the rest was stored in a freezer at −80 °C until before analysis.

### 2.3. Chemical Analysis

#### 2.3.1. UPLC-Q-Orbitrap MS/MS Analysis

The UPLC-Q-Orbitrap-MS system was equipped with a Q Exactive™ hybrid quadrupole-Orbitrap mass spectrometer (Thermo Fisher Scientific, San Jose, CA, USA) and a Waters I Class ACQUITY™ UPLC system (Waters Corporation, Milford, MA, USA). The chromatographic separation was achieved on an ACQUITY UPLC^®^ BEH C18 (2.1 × 100 mm, 1.7 µm, Waters, Washington, DC, USA) protected by an ACQUITY UPLC^®^ BEH C18 VanGuard™ pre-column (2.1 × 5 mm, 1.7 µm, Waters, USA) at 30 °C. Through the optimization of sample extraction time and extraction solvent, it was finally determined to dissolve 2.5 g of the sample into a 25 mL volumetric flask with methanol to the scale, followed by sonication for 30 min and centrifugation at 14,000 rpm for 10 min. The supernatant was collected and filtered with a 0.22 µm membrane. The mobile phase was 0.1% *v/v* formic acid aqueous solution (A) and acetonitrile solution (B). The elution gradient was as follows: 0–3 min, 95%A–93%A; 3–5 min, 93%A–80%A; 5–8 min, 80%A–76%A; 8–9 min, 76%A–30%A; 9–11 min, 30%A–24%A; 11–12 min, 24%A–22%A; 12–16 min, 22%A–5%A. Efficient and symmetrical peaks were obtained at a flow rate of 0.3 mL/min with a sample injection volume of 1 µL.

The Q-Orbitrap mass spectrometer was operated in positive and negative ion mode with a HESI source. The HESI source parameters were set as follows: sheath gas flow rate, 35 arb; auxiliary gas flow rate, 8 arb; spray voltage, 3.0 kV; capillary temperature, 350 °C; and auxiliary gas heater temperature, 350 °C. The full MS/dd-MS2 scanning method was applied and the full scanning range was m/z 67~1000 in the qualitative experiment on mulberries. The normalized collision energy (NCE) was optimized to 20/30/50 V. Xcalibur 4.1 (Thermo Fisher Scientific) was used for instrument control, data acquisition, and data analysis. Then the collected original data were imported into Compound Discoverer™ 3.3 (Thermo Fisher Scientific) and a workflow was established for data processing; the main operations were as follows: conduct peak alignment, peak extraction, the mass loss to be less than 5 ppm. Database selection included the Chemspider online database and mzVault local database for compound prediction. Finally, identification was conducted based on the score of compound matching, retention time, peak area, and comparison of references.

#### 2.3.2. HS-SPME-GC-MS Analysis

HS-SPME-GC-MS is a headspace autosampler system consisting of an Agilent 7890B-7000D gas chromatography and mass spectrometry detector (Agilent, Thermo fisher, CA, USA) combined with an SPME fiber (supelco, Bellefonte, Penn.). Mulberries were crushed by domestic juicer (WBL2501B, Midea Group Co., Ltd., Guangdong, China) and 2.5 g was weighed accurately into a 20 mL glass sampling vial, then the headspace bottle was sealed with a screw cap with a silicon gasket. The samples were incubated at 50 °C for 40 min and then 500 µL of each headspace sample was automatically inserted by a DVB/CAR/PDMS SPME needle (2 cm, 50/30 µm; supelco) at 240 °C. The chromatographic column was a DB-WAX capillary column (30 m × 0.25 mm ID, 0.25 μm, 122-7032, J&W Scientific, Folsom, CA, USA) at 40 °C. Helium (99.999%) was used as the carrier gas at a flow rate of 1 mL/min and the column temperature was as follows: initially held at 40 °C for 3 min, then increased to 140 °C at a rate of 4 °C/min, finally heated to 220 °C at 10 °C/min. The injector temperature was maintained at 250 °C with a splitless inlet. When the ionization energy was 70 ev, the mass spectrometry worked in electron ionization (EI) mode. The forward sample port and ion source temperature were set at 250 °C and 230 °C, respectively; the resolution time was 5 min, and the quadrupole temperature was 150 °C. Finally, the scanning range was set to m/z 30~650. All samples were subjected to 5 repeated experiments. The spectrum peak was searched by the NIST.17 standard mass spectrometry library (Agilent, CA, USA) of the chemical workstation for qualitative analysis of unknown compounds, while the retention index (RI) of each compound was calculated using N-alkanes C8–C20 as the external reference.

#### 2.3.3. HS-GC-IMS Analysis

Samples were analyzed on the HS-GC-IMS apparatus (Flavourspec^®^, G.A.S, Dortmund, Germany) with an autosampler (CTC Analytics AG, Switzerland) equipped with a 1 mL heated airtight syringe. The chromatographic column was an FS-SE-54-CB-1 (CS-Chromatographie Service GmbH, Germany) capillary column (15 m × 0.53 mm ID, 0.5 μm). Sample pretreatment was the same as for HS-SPME-GC-MS; ground mulberry samples (2.5 g) were weighed accurately into a 20 mL glass sampling vial then sealed. The samples were incubated at 80 °C for 20 min at 250 rpm and then 500 µL of each headspace sample was automatically injected by a heated syringe at 80 °C into the injector (85 °C, splitless mode). The temperature of the capillary column (15 m × 0.53 mm ID, 1 µm, Beijing, China) was 60 °C. Nitrogen (99.999%) was used as a carrier gas and its flow rate was first set at 2 mL/min for 2 min, then increased to 10 mL/min over 10 min, and increased to 150 mL/min over 20 min and held for 10 min. The resulting ions were driven to a drift tube (9.8 cm in length), which was operated at a constant temperature (45 °C) and voltage (5 kV). The flow rate of the drift gas (N2) was set at 150 mL/min. All analyses were conducted 5 times. N-ketone C4-C9 standard mix was used to calculate the retention index (RI) of each compound as the external reference. HS-GC-IMS was used to support analysis software for data processing and related graphic generation. VOCal software was used for visualizing chromatograms and data analysis, with built-in NIST and IMS databases for qualitative analysis of substances.

### 2.4. Antioxidant Activity

The antioxidant capacity of morula was characterized by DPPH radical scavenging ability. The preparation method of mulberry fruit extract was modified as follows [24]: 1 g of lyophilized mulberry powder was accurately weighed and extracted by reflux with 70% ethanol at the ratio of 1:20 for 1 h. After the filter residue was re-extracted, the two filtrates were combined, then scaled into a 100 mL volumetric flask. The extracting solution was centrifuged at 10,000 rpm for 10 min (20 °C) and the supernatant was collected. The extracts of four maturation stages were formulated into test solutions of 0.1, 0.2, 0.4, 0.6, 0.8, 1, 1.5, 2, 2.5 mg/mL to determine the free radical scavenging activity [25]. The absorbance measured at 517 nm after reacting 2 mL of samples with different concentrations and 2 mL of DPPH solution (0.2 mM) in a dark environment for 30 min was recorded as A_1_, and the absorbance measured after reacting 2 mL of samples with 2 mL of anhydrous ethanol under the same conditions was recorded as A_2_. After reacting 2 mL of DPPH with 2 mL of anhydrous ethanol in the dark for 30 min, the absorbance was recorded as A_3_. Three parallel experiments were performed for each sample and the absorbance was measured at 517 nm using an Alpha-1502 spectrophotometer (Shanghai Spectral Yuan Instrument Co., Ltd., Shanghai, China). Test solutions of vitamin C were prepared identically and absorbance was determined. The free radical scavenging rate of DPPH was calculated following the formula.
DPPH scavenging effect (%) =(A_3_ + A_2_ − A_1_)/A_3_ × 100%

## 3. Results and Discussion

### 3.1. UPLC-Q-Orbitrap MS/MS Analysis

In order to achieve better resolution for as many organic compounds as possible, the parameters of UPLC were optimized. The ACQUITY UPLC^®^ BEH C18 column, HSS T3 column, and HSS C18 column were evaluated by the chromatographic peak shape and the number of peaks (Appendix A). The BEH C18 had a higher peak resolution and greater abundance in chromatograms by comparison. Different mobile phases (water–methanol/water–acetonitrile/0.1% *v*/*v* aqueous formic acid–methanol/0.1% *v*/*v* aqueous formic acid–acetonitrile) also had an effect on the chromatographic peak shape and the number of peaks. It was found that when 0.1% *v*/*v* aqueous formic acid–acetonitrile was used as the mobile phase, the peak shape was better. For different column temperatures, 20 °C, 25 °C, 30 °C, 35 °C, 40 °C, and 45 °C were compared and 30 °C was found to be the most suitable (Appendix A).

In addition, the key parameters of Q-Orbitrap were also optimized, including spray voltage, capillary temperature, and auxiliary gas heater temperature. Considering the diversity of sample constituents, three compounds were selected as references in positive and negative ion modes, respectively. Isoquercetin, chlorogenic acid, and astragaloside were selected in positive ion mode; rutin, quercetin, and protocatechuic acid were selected in negative ion mode. Appendix A is the presentation of experimental results. For spray voltage, 2.0 kV and 3.0 kV did not differ much, but 3.5 kV was less stable. The effect of capillary temperature was found to increase gradually with increasing temperature, but the peak areas of individual constituents decreased significantly when raised to 400 °C, leaving 350 °C as the final option. The auxiliary gas heater temperature also basically showed an upward trend, but the peak area was largest at 350 °C for components with less content, such as violacein and quercetin. The spray voltage was set to 3.0 kV and the capillary temperature and auxiliary gas heater temperature were simultaneously set to 350 °C for all samples to be analyzed.

After sample injections were repeated five times for data acquisition, Compound Discoverer™ 3.3 (Thermo Fisher Scientific) was used as qualitative analysis software matching the Q-Orbitrap, through an online database (mzCloud) and a local database (mzVault) for the preliminary prediction of compounds. Compound Discoverer™-screened compounds could be further analyzed by means of ion peak retention time contrast, mass-to-charge ratio, and secondary characteristic ion fragmentation using Xcalibur 4.1 software (Thermo Fisher Scientific). Finally, combining the literature and standard information, 68 n-VOCs were identified (Table 1). 29 n-VOCs were identified in positive ion mode and 39 in negative, which were mainly components of flavonoids, phenolic acids, and so on.

### 3.2. HS-SPME-GC-MS Analysis

Key parameters were optimized before the experiments formally started. Different extraction fiber head sorption membranes had large differences in the sorption capacity of VOCs, the variation of extraction temperature and extraction time also had effects on the analysis of sample components. Firstly, the samples were analyzed by using four different extraction fiber heads, including Poly, PDMS, PDMS/DVB and DVB/CAR/PDMS. The total ion chromatogram (TIC) (Appendix A) showed that the chemical components analyzed by the DVB/CAR/PDMS probe were the most abundant and the number of chromatographic peaks was also the largest. Secondly, the samples were incubated at 40 °C, 50 °C, 60 °C, and 70 °C, respectively. Analysis of chromatograms (Appendix A) revealed that the peak area and number of peaks were optimal at an incubation temperature of 50 °C, as the temperature continued to increase to 60 °C when the peak area and number of peaks decreased significantly. Finally, the extraction time was optimized. The samples were extracted for 20 min, 30 min, 40 min, and 50 min then contrasted with the TIC (Appendix A). The peak area of 30 min was significantly larger than that of 20 min, but the peak areas of the chromatographic peaks did not change significantly after the subsequent increase times of 40 min and 50 min. Thus, 30 min of extraction time was chosen as the most suitable. To sum up, the VOCs extracted by incubation with a DVB/CAR/PDMS needle for 30 min at 50 °C were the most abundant.

The TIC of mulberry samples in different stages is shown in Figure 2. I to IV were mulberry fruits that grew from immature to mature. Some different VOCs are marked in Figure 2. It can be seen from the figure that the difference was not only in the content of VOCs, but also in the type of VOCs. A total of 64 compounds were identified according to the retention index, matching value, and literature reference in Table 2. It is worth noting that 16 kinds of VOCs (hexanal; n-hendecane; p-xylene; dodecane; hex-2-enal; trans-2-hexenal; 2-pentylfuran; p-cymene; 1, 2, 4-trimethylbenzene; n-tridecane; 1-nonanal; 5-ethyl-1-cyclopentene-1-carboxaldehyde; (E)-2-octenal; 1-octen-3-ol; β-cyclocitral and phenylacetaldehyde) existed in the whole process of mulberry growth, 14 kinds (2-ethyl toluene; m-cymene; cyclohexene; 2-heptenal; 6-methyl-5-hepten-2-one; α, p-dimethylstyrene; acetic acid; trans, trans-2, 4-heptadienal; linalool; (−)-isocaryophyllene; cis-5-octen-1-ol; α-farnesene and jasmone) existed in stage I, seven kinds (valeraldehyde, o-cymene, decyl aldehyde, n-pentadecane, 3-heptylacrolein, decyl alcohol and phenethyl alcohol) existed in stage II, one kind (irisone) existed in stage III, and two kinds ((2E, 4Z)-decadienal and hexanoic acid) existed in stage IV. The remaining VOCs existed in several of them. Moreover, it could be clearly seen that VOCs were gradually decreasing with the maturity of the mulberry fruit from the identified VOCs.

### 3.3. HS-GC-IMS Analysis

The extraction and analysis of the GC-IMS data were performed with a Laboratory Analytical Viewer (LAV) (version 2.2.1, G.A.S, Dortmund, Germany). VOCs were identified based on the IMS database of the GC-IMS Library Search application software. The RI of each compound was calculated using N-ketones C4–C9 as external references. Sample information of different periods of HS-GC-IMS analysis was displayed in the form of a topographic map and fingerprint in Figure 3. VOCs extracted from the samples were displayed in the spectrum of Figure 3A. The red vertical line at abscissa 1.0 of the figure was the normalized reaction ion peak (RIP). Each point on the right of the RIP represented a VOC in the sample. The redder the point, the higher the concentration of this VOC. Therefore, the dynamic changes of VOCs in mulberry fruits during different growth periods were visually displayed by topographic maps. To further compare the difference of components in different mature stages, the fingerprints of 12 samples (three in each stage) are shown in Figure 3B. Each row of the fingerprint represents a sample and each column represents a VOC. The reason for producing monomers or dimers was that monomer ions and neutral molecules may form attached substances in the drift zone, which makes single compounds produce multiple signals. The VOCs mainly included aldehydes, esters, and ketones, while hydrocarbons and heterocycles were less frequent. The fingerprint spectrum (Figure 3B) showed obvious differences in VOCs at different maturation stages. The VOCs in region A that were constantly present during maturation included monomers of nonanal, α-phellandrene, heptanal, hexanal, 2-methylpropanal, and methyl benzoate; and dimers of cyclohexanone, (E)-2-hexen-1-ol, ethyl acetate, hexanal, and 2-methylpropanal. The concentration of VOCs in region B (monomers of 1.8-cineole, α-phellandrene, 2-pentyl furan, hexanal, 2-acetylfuran, and ethyl pyrazine; and dimers of 2-heptanone, (E)-2-octenal, (E, E)-2,4-heptadienal and 3-heptanol) gradually decreased during the maturation of the mulberry. However, VOCs in region E were on the contrary; for example, the concentration of (E)-2-pentenal monomer gradually increased with maturation. VOCs in region C, including monomers of 2-heptanone, 3-methylbutanal, (E)-hept-2-enal, oct-1-en-3-ol, phenylacetaldehyde, 1-octen-3-one, (E)-2-octenal, (E, E)-2,4-heptadienal, and 2,4-heptadienal monomer; and dimers of 1-octen-3-one, (E)-2-pentenal and (E)-hept-2-enal, disappeared suddenly during the progression from stages I and II to stages III and IV, and could be used as markers for the transition of mulberry fruits from immature to mature. VOCs in region D peaked in phase II, as maturation gradually disappeared, and they were borneol monomer, α-pinene monomer, β-ocimene monomer, benzyl acetate monomer, 2-ethyl-3,5-dimethyl pyrazine monomer, and ethyl pyrazine monomer. In region F were the most abundant VOCs in stage IV, which are the main characteristic compounds in the ripening stage, including dimers of benzaldehyde, benzene acetaldehyde, nonanal, heptanal, and 3-methylbutanal. A total of 34 VOCs were identified through the GC-IMS database in all samples, and 16 VOCs with both presented monomers and dimers are shown in Table 3.

### 3.4. Antioxidant Activity

In recent years, one of the most popular trends in food-based health products and medical research is the use of natural bioactive substances from plants as a source of antioxidant and anti-aging chemicals [26]. The antioxidant activity of mulberries could be indicated by the free radical scavenging rate of DPPH. Mulberries at four different maturation stages were assayed in triplicate to obtain experimental data (Appendix A). The trend of antioxidant activity variation of samples at various maturation stages is displayed in Figure 4B. In the experimental concentration range (0.1~2.5 mg/mL), the radical scavenging rate of DPPH exhibited some degree of dose dependence. A lower half-inhibitory concentration (IC50) indicates a higher free radical scavenging rate. According to the experimental data, the IC50 values from Phase I to Phase IV were 0.36, 0.41, 0.33, and 0.22 mg/mL, respectively (Figure 4A). The results indicated that the antioxidant capacity of mature mulberry fruits (stage IV) was the most outstanding. The results were consistent with the laws of food sales markets, where dried morula and morula beverages are adopted using ripe fruits.

### 3.5. Comprehensive Analysis

A total of 166 compounds were identified by UPLC-Q-Orbitrap-MS analysis, HS-SPME-GC-MS analysis, and HS-GC-IMS analysis. The peak areas were taken as a reference for statistical analysis of the collected sample data. Firstly, PCA was performed using Origin 2022 software. PCA, as an unsupervised multivariate statistical analysis method, can reduce the dimensionality of data and reveal the regularity and difference between samples by comparing principal component factors. A, B and C in Figure 5 represented the results of UPLC-Q-Orbitrap-MS analysis, HS-SPME-GC-MS analysis, and HS-GC-IMS analysis, respectively. The samples in the four stages achieved clustering in all three means of analysis, thereby illustrating distinct differences in organic matter composition across stages.

RFA was performed using metaboanalyst 5.0 (https://www.metaboanalyst.ca/ (accessed on 11 July 2023)). RFA is a supervised statistical analysis method that does not require dimensionality reduction of the data because it can handle data in very high dimensions. No feature selection was performed at the same time because feature subsets were randomly selected. Insensitivity to missing values was one of its advantages. Accuracy was maintained even when a significant fraction of the features were missing. The characteristic components with an RFA score value higher than 0.018 were selected for analysis in the three analyses. It is noteworthy that in the UPLC-Q-Orbitrap-MS analysis (Figure 5D), cyanidin-3-rutoside, cyanidin-3-glucoside, and catechin were the characteristic components in stage I, and sangenol D was the most specific in stage II. In the HS-SPME-GC-MS analysis (Figure 5E) and the HS-GC-IMS analysis (Figure 5F), 2-geptenal, (E)-2-octenal, jasmone, linalool, and acetyl pyrazine monomer were the characteristic components of phase I; n-tridecane and 2-ethyl-3,5-dimethylpyrazine monomer were the characteristic components of phase II; β-cyclocitral, 1-nonanal, ethyl acetate, and dodecane were the characteristic components of phase III; and trans-2-nonenal, isopropyl alcohol monomer, and heptanal dimer were the characteristic components of phase IV.

In addition, the differential component contents were also analyzed. OPLS-DA models were first built on three datasets (UPLC-Q-Orbitrap-MS analysis, HS-SPME-GC-MS analysis, and HS-GC-IMS analysis) of the samples using SIMCA 14.1 software. OPLS-DA is a supervised method that can only identify the information difference of different classes of samples and eliminate the influence of irrelevant data. According to the OPLS-DA model, organic compounds with VIP ≥ 1 were selected, and peak area was used as a reference for component content. Peak area normalization was performed, and the selected components were subjected to relative quantitative analysis. The panels A, B, C of Figure 6 present their percentages in each period. The proportions of VOCs in immature mulberries (stages I and II) were relatively large, such as 3-methyl-6-(1-methylethylidene) cyclohexene, (−)-isocaryophyllene, and m-cymene, which only existed in the immature stage. The proportion of trans-2-hexenal, (E)-2-octenal, and 5-ethyl-1-cyclopentene-1-carboxaldehyde was greater than 80% (Figure 6B); The proportion of (E)-2-octenal dimer and 2-pentylfuran monomer was 94%, while the relative content of (E)-hept-2-enal dimer, (E)-2-octenal monomer, (E)-hept-2-enal dimer, and phenylacetaldehyde monomer was ≥75% (Figure 6C). However, the mature mulberries (stages III and IV) had fewer VOCs, mainly phenolic acids and flavonoids. The proportion of cyanidin-3-rutoside was 95% and crotonic acid was 79% in maturing mulberries, in Figure 6A. These mentioned compounds had a relative content ≥ 75% in a certain period, and it was reasonable to use these as biomarkers for the different growth periods of the mulberry.

It was found that some compounds changed with the maturation of the mulberry. For non-volatile components, flavonoids were the dominant phenolic compounds in mulberry, and flavonoids are known to be cytotoxic to human cancer cell lines [8]. Cyanidin-3-rutoside, cyanidin-3-glucoside, and catechin all belong to this class of compounds. For the volatile components, alcohols, aldehydes, ketones, and olefins were the main aroma components in the mulberry. For example, linalool, on account of its protective effects and low toxicity, can be used as an adjuvant of anticancer drugs or antibiotics. Therefore, linalool has a great potential to be applied as a natural and safe alternative therapeutic [27]. In addition, the mulberry also has good development prospects in food development, especially the black mulberry. Studies had shown that the content of total phenols, total flavonoids, total Anthocyanidin, and antioxidant compounds in the black mulberry are higher than those in the red mulberry or white mulberry [28]. Due to the rich nutritional value of the mulberry, it had been widely used in the production of modern food and dietary plans. It can not only be directly used after air drying, but can also be processed into wine, syrup, canned food, fruit juice, jam, and beverages [29]. Although the mulberry has been widely researched and developed, there are still some issues that need further research. During the processing of mulberries, functional components may be lost. In view of the poor taste of fresh mulberry food, if the processing technology can be improved such that little or none of the Active ingredient of processed products is lost, then the processing and utilization of mulberries could be very beneficial. How to establish a storage and preservation system from the field to consumers, extend the shelf life of mulberry trees, maintain their freshness, and improve their economic value are also urgent problems to be solved. The characterization of mulberry ingredients in this work provide a basis for in-depth research on the flavor substances of mulberry in the future and for improving the quality of its fresh food.

## 4. Conclusions

The n-VOCs and VOCs of mulberries during different growth periods were determined by UPLC-Q-Orbitrap-MS, HS-SPME-GC-MS, and HS-GC-IMS. A total of 166 compounds were identified, including 68 n-VOCs and 98 VOCs. A total of 13 kinds were identified simultaneously in both the HS-SPME-GC-MS and HS-GC-IMS analyses. The n-VOCs were mainly flavonoids and phenolic acids, while the VOCs mainly included aldehydes and esters. PCA showed that mulberries could achieve obvious clustering at different growth periods. The RFA and OPLS-DA were further adopted to verify PCA and evidenced that the result was reasonable. Quantitatively, 3-methyl-6-(1-methylethylidene)cyclohexene, (−)-isocaryophyllene, and m-cymene were displayed in relatively high amounts in stage I; neochlorogenic acid, delphinidin 3,5-diglucoside, and n-tridecane were shown in relatively high amounts in stage II; (E)-2-octenal dimer and 2-pentylfuran monomer were presented in relatively high amounts in stage I and stage II; 3-hydroxy-3-(methoxycarbonyl) pentanedioic acid, α-phellandrene dimer, and ethyl acetate dimer were presented in relatively high amounts in stage III; and cyanidin-3-rutoside, crotonic acid, phenylacetaldehyde, and 3-methylbutanal dimer were presented in relatively high amounts in stage IV. Additionally, antioxidant experiments revealed that as the mulberries continued to mature, flavonoids and phenolic acids continued to increase, and antioxidant activity gradually increased. Stage IV had the highest antioxidant capacity. Among the identified VOCs, 24 compounds were presented in the growth full process, and 74 existed with the emergence and disappearance of dynamic changes. Conclusively, an effective strategy has been established for comprehensively characterizing the composition of mulberry fruits, presenting a dynamic analysis of sample composition and differences at different stages. Annotating the dynamic changes of ingredients during the growth process helps to achieve quality analysis, develop medicinal and edible value, and provide a basis for food safety and authenticity verification.

## Figures and Tables

**Figure 1 foods-12-03335-f001:**
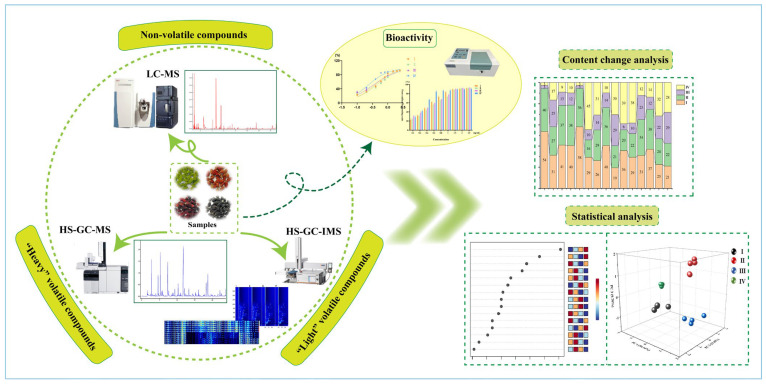
Experimental strategy.

**Figure 2 foods-12-03335-f002:**
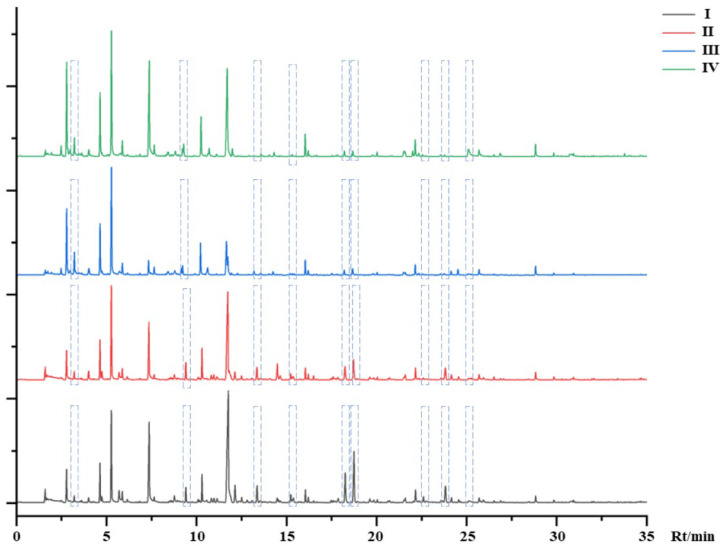
The total ion chromatogram (TIC) of HS-SPME-GC-MS.

**Figure 3 foods-12-03335-f003:**
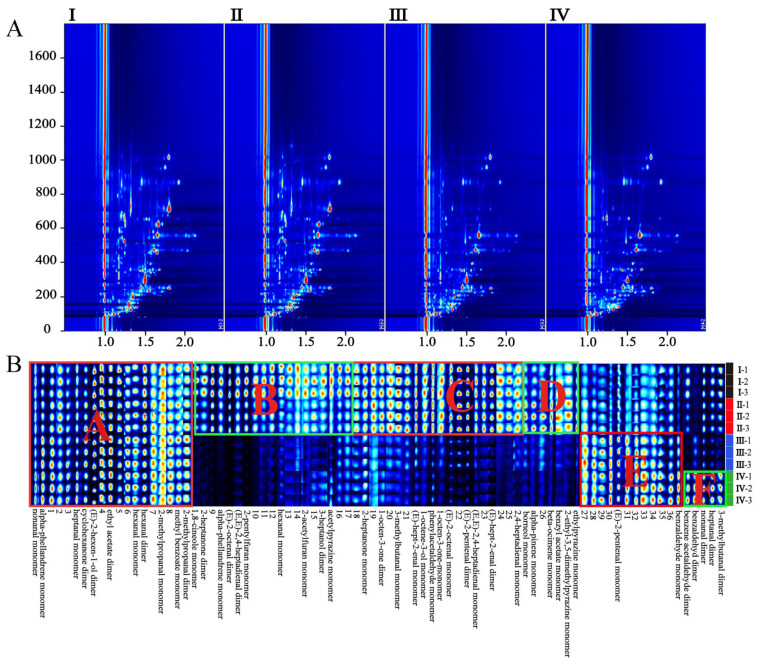
(**A**): 2D top views of mulberry samples detected by HS-GC-IMS; (**B**): the fingerprints of 12 samples (three in each stage); I–IV are samples of mulberry at different mature stages. The ABCDEF region in (**B**) is a division of differential components, which is detailed in the text.

**Figure 4 foods-12-03335-f004:**
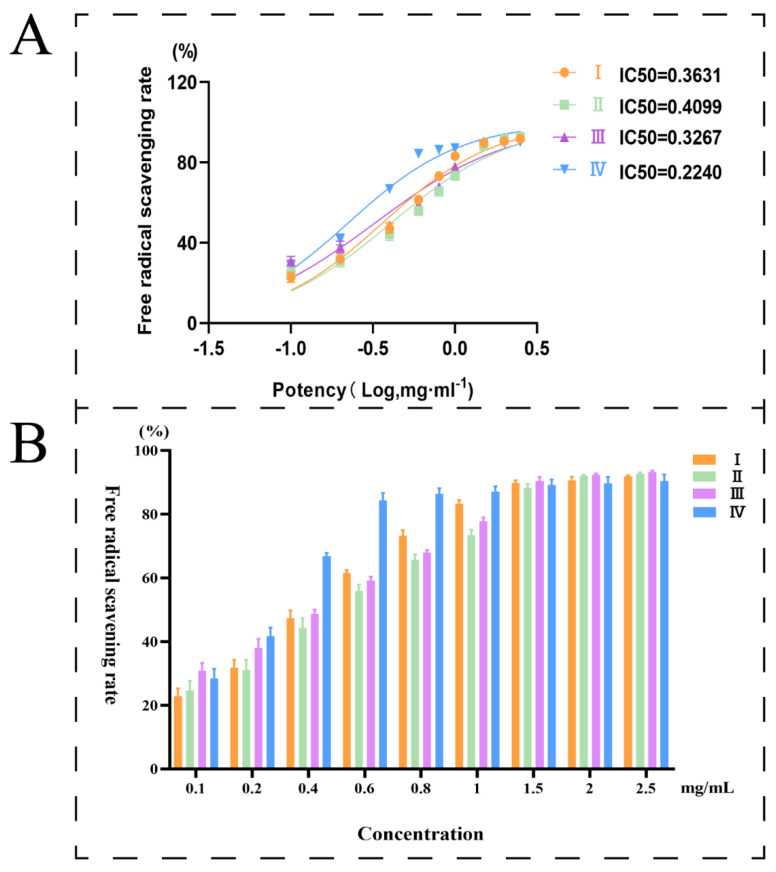
(**A**): Free radical scavenging rate of DPPH; (**B**): IC50 for different stages.

**Figure 5 foods-12-03335-f005:**
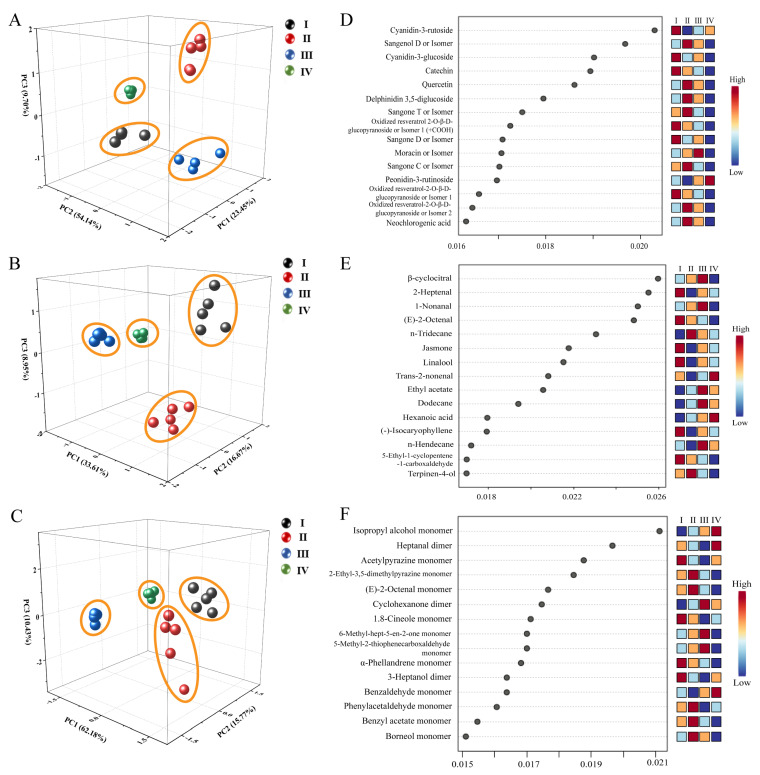
(**A**–**C**): PCA analysis by UPLC-Q-Orbitrap-MS, HS-SPME-GC-MS, and HS-GC-IMS, respectively; (**D**–**F**): RFA by UPLC-Q-Orbitrap-MS, HS-SPME-GC-MS, and HS-GC-IMS, respectively.

**Figure 6 foods-12-03335-f006:**
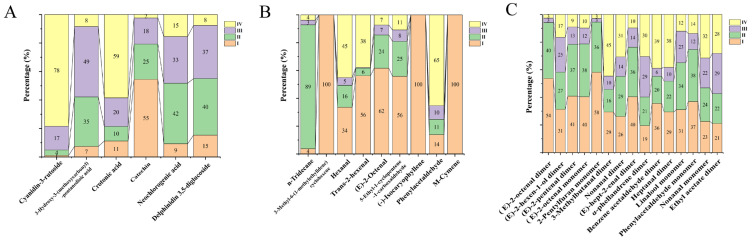
(**A**–**C**): Percentage of component content measured by UPLC-Q-Orbitrap-MS, HS-SPME-GC-MS, and HS-GC-IMS, respectively.

**Table 1 foods-12-03335-t001:** Information about the qualitative compounds detected by UPLC-Q-Orbitrap MS.

No.	Rt/min	Formula	Ion Type	Detected Mass (m/z)	ppm	MS/MS Fragments	Identification
1	0.96	C_4_H_6_O_2_	[M + H]+	87.04408	0.31	69.03360	Crotonic acid
2	1.09	C_6_H_8_O_7_	[M − H]−	191.01857	−0.28	179.05521, 161.04453, 111.00720, 147.02855	Isocitric acid
3	1.32	C_6_H_8_O_7_	[M − H]−	191.01848	−0.76	111.00717, 147.02855, 129.01785, 85.02780	Citric acid
4 *	1.87	C_7_H_6_O_5_	[M − H]−	169.01337	1.28	157.01324, 134.01625	Gallic acid
5	2.15	C_7_H_10_O_7_	[M − H]−	205.03438	0.48	173.00821, 111.00729	3-Hydroxy-3-(methoxycarbonyl) pentanedioic acid
6 *	3.48	C_7_H_6_O_4_	[M − H]−	153.01820	−0.21	109.02804	Protocatechuic acid
7	4.13	C_16_H_18_O_9_	[M − H]−	353.08737	−1.88	191.01785, 163.03912	1-Caffeioylquinic acid
8	4.35	C_21_H_22_O_11_	[M + H]+	451.12396	1.05	289.07117, 153.09128	Oxidized resveratrol 2-O-β-D-glucopyranoside or Isomer 1 (+COOH)
9	4.94	C_20_H_24_O_13_	[M + H]+	473.12946	1.03	341.08725, 179.07043	5-Hydroxycoumarin-7-O-[β-D-furosyl-(1→6)-O-β-D-glucopyranoside
10	5.04	C_21_H_26_O_13_	[M + H]+	487.14505	0.89	451.11154, 341.08774, 179.06096	7-Hydroxycoumarin-6-O-α-L-rhamnosyl-(1→6)-O-β-D-pyran-Glucoside
11	5.21	C_20_H_22_O_9_	[M + H]+	407.13403	0.92	245.08116, 227.12811	Oxidized resveratrol 2-O-β-D-glucopyranoside or Isomer 2 (+COOH)
12	5.41	C_21_H_20_O_11_	[M + H]+	449.10803	0.43	287.05411, 181.04961	Kaempferol-7-O-glucopyranoside
13 *	5.43	C_16_H_18_O_9_	[M + H]+	355.10233	−0.34	287.05411, 135.04408, 117.03352, 89.03857	Chlorogenic acid
14	5.46	C15H14O6	[M − H]−	289.17175	−0.89	271.02234, 245.08195, 109.02795	Catechin
15	5.48	C_27_H_30_O_15_	[M + H]+	595.16589	0.25	449.10828, 287.05429	Cyanidin-3-rutoside
16	5.51	C_27_H_30_O_17_	[M + H]+	627.15552	−0.08	465.10452, 303.05020	Delphinidin 3,5-diglucoside
17	5.59	C_16_H_18_O_9_	[M − H]−	353.08707	1.01	173.04445, 135.04370, 111.04362, 93.03295	Neochlorogenic acid
18	5.64	C_27_H_32_O_16_	[M + H]+	613.17639	0.14	431.15253, 305.06586, 137.02333	Dihydroquercetin-7-rutinoside
19	5.71	C_21_H_20_O_10_	[M + H]+	433.11343	1.18	271.05957, 377.08466	Pelargonidin-3-glucoside
20	5.71	C_27_H_30_O_14_	[M + H]+	579.17114	0.54	433.11343, 377.08466, 271.05957	Pelargonidin-3-rutinoside
21 *	5.86	C_9_H_8_O_4_	[M − H]−	179.03394	0.28	135.04385, 121.02808	Caffeic acid
22	6.01	C_33_H_40_O_21_	[M + H]+	773.21338	1.61	611.16083, 465.10391, 303.05032	Quercetin methyl pentoside dihexoside
23	6.23	C_33_H_40_O_20_	[M − H]−	755.20636	6.34	301.03577, 489.06564	Quercetin-3-O-α-Rhamnose-β-Glucose-α-Rhamnoside
24	6.39	C_20_H_22_O_9_	[M + H]+	407.13443	−1.39	227.09143, 245.06284	Oxidized resveratrol-2-O-β-D-glucopyranoside or Isomer 1
25	6.42	C_15_H_10_O_6_	[M + H]+	287.05548	1.62	259.06143, 241.05542, 153.01125	Kaempferol
26	6.56	C_21_H_20_O_11_	[M + H]+	449.10840	1.25	423.06052, 287.05533	Cyanidin-3-galactoside
27	6.58	C_20_H_22_O_9_	[M + H]+	407.13342	−0.57	227.09189, 245.00316	Oxidized resveratrol-2-O-β-D-glucopyranoside or Isomer 2
28 *	6.73	C_27_H_30_O_16_	[M + H]+	611.16071	0.08	465.10327, 303.05005	Rutin
29	6.94	C_21_H_20_O_11_	[M + H]+	449.10818	0.77	287.05536	Cyanidin-3-glucoside
30	6.97	C_27_H_30_O_15_	[M + H]+	595.16595	0.35	449.10442, 287.05221	Cyanidin-3-rutinoside
31 *	7.04	C_21_H_20_O_12_	[M + H]+	465.10312	0.79	303.04993	Quercetin-3-O-glucoside
32	7.35	C_27_H_30_O_15_	[M + H]+	595.16959	0.35	617.14771, 449.10831, 287.05530	Kaempferol-3-O-rutoside
33*	7.46	C_15_H_12_O_7_	[M−H]−	303.05121	−0.2	285.04056, 255.03001	Taxifolin
34	7.55	C_25_H_24_O_12_	[M + H]+	517.13434	0.55	492.31717, 355.10272, 137.13260	4, 5-Dicaffeoylquinic acid
35 *	7.76	C_21_H_20_O_11_	[M + H]+	449.10834	1.11	423.09018, 287.05536	Astragaloside
36	7.83	C_25_H_24_O_12_	[M + H]+	517.13452	0.91	499.12384, 355.10272, 287.05527, 163.03908	1, 5-Dicaffeoylquinic acid
37	8.35	C_25_H_24_O_12_	[M + H]+	517.1347	1.26	499.12396, 355.10294, 287.05518, 163.03917	1, 3-Dicaffeoylquinic acid
38	8.47	C_23_H_22_O_12_	[M − H]−	489.10480	0.41	455.11292, 269.00580	Kaempferol-3-O-6”-acetylglucoside
39	8.72	C_30_H_32_O_6_	[M − H]−	487.21893	0.43	243.12364	Sangenol J or Isomer 1
40	8.97	C_30_H_32_O_6_	[M − H]−	487.21841	−0.64	349.15097, 243.12389	Sangenol B or Isomer 1
41	9.07	C_30_H_36_O_6_	[M − H]−	491.24063	1.02	489.23305, 151.03879	Sangenol C or Isomer 1
42	9.46	C_30_H_32_O_6_	[M − H]−	487.21893	0.43	349.01346, 231.12369	Sangenol D or Isomer
43	9.53	C_45_H_44_O_11_	[M − H]−	759.28534	−6.65	693.24182, 581.32056, 353.08856	Morin O or Isomer
44	9.56	C_30_H_32_O_6_	[M − H]−	487.2197	−0.75	349.15085, 243.12381, 231.08730	Sangenol J or Isomer 2
45	9.63	C_45_H_42_O_11_	[M − H]−	757.26642	−0.11	647.27258	Sangone W
46 *	9.67	C_15_H_10_O_7_	[M − H]−	301.03525	−1.22	300.02280, 285.04086, 276.99191, 151.03889	Quercetin
47	9.70	C_25_H_26_O_6_	[M − H]−	421.15134	0.43	401.16150, 363.02887, 349.13083, 309.01758	Sangone T or Isomer
48	9.74	C_30_H_32_O_6_	[M − H]−	487.21964	0.82	349.18784, 243.12370, 231.12364	Sangenol B or Isomer 2
49	10.09	C_40_H_36_O_11_	[M + H]+	693.23053	−6.28	513.25769, 421.27188, 355.11783, 299.12848	Sangone K
50 *	10.19	C_40_H_36_O_11_	[M + H]+	693.23209	−6.63	671.46423, 513.25549, 473.22809, 299.10492	Sangone G
51	10.21	C_25_H_26_O_7_	[M − H]−	437.16071	−0.25	349.00272, 281.00177, 125.02299	Cyclomoritol or Isomer
52	10.34	C_25_H_26_O_7_	[M − H]−	437.16107	−0.58	349.00919, 281.00211	Morusinol or Isomer
53	10.41	C_25_H_26_O_6_	[M − H]−	421.16620	−0.5	401.08832, 349.00931, 311.01199, 309.17148	Sangone D or Isomer
54	10.47	C_30_H_36_O_6_	[M − H]−	491.24423	−0.87	365.00418, 313.23911, 151.03887	Morganone C or Isomer 2
55	10.54	C_30_H_34_O_6_	[M − H]−	489.22943	0.85	349.00934, 309.20761, 243.00668, 151.03873	Morganone C or Isomer
56	10.58	C_25_H_26_O_7_	[M − H]−	437.16071	−0.25	349.00919, 281.00195, 125.02296	Sangone U or Isomer 1
57	10.68	C_25_H_26_O_7_	[M − H]−	437.16138	0.11	349.00919, 281.00177, 125.02294	Sangone U or Isomer 2
58	10.86	C_25_H_24_O_6_	[M − H]−	419.15039	0.33	375.19473, 349.00906, 227.06589, 147.02870	Moracin or Isomer 1
59	10.96	C_25_H_26_O_6_	[M − H]−	421.16632	−0.21	350.01184, 309.20764, 269.13925, 231.00548	Sangone F or Isomer
60	10.99	C_25_H_24_O_6_	[M − H]−	419.15039	0.33	349.01108, 309.20758, 227.06552, 173.05609	Moracin or Isomer 2
61	11.27	C_25_H_26_O_6_	[M − H]−	421.16534	0.64	351.07144, 334.99438, 311.22174, 297.23419	Sangone C or Isomer
62	11.48	C_25_H_24_O_6_	[M − H]−	419.15039	0.33	349.01154, 227.06543, 173.04459, 147.02866	Moracin or Isomer 3
63	11.65	C_30_H_36_O_6_	[M − H]−	491.24490	0.48	365.00574, 151.03876	Morganone C or Isomer 3
64	12.32	C_25_H_22_O_6_	[M − H]−	417.13577	−0.41	365.27014, 334.24670, 289.00955, 241.10832	Cyclomulberrin
65	13.08	C_28_H_32_O_15_	[M + H]+	609.18646	0.46	607.25531, 301.15366	Peonidin-3-rutinoside
66	13.41	C_25_H_24_O_6_	[M − H]−	419.15039	0.33	349.01041, 217.02921, 173.08087	Moracin or Isomer 4
67	13.69	C_30_H_36_O_6_	[M − H]−	491.24063	1.02	473.28308, 313.23923	Morganone C or Isomer 4
68	13.80	C_30_H_34_O_6_	[M − H]−	489.22742	0.51	309.04639, 255.00645, 243.19635	Morganone B or Isomer

*: Verified by standard materials.

**Table 2 foods-12-03335-t002:** Information about the qualitative VOCs detected by HS-SPME-GC-MS.

No.	Rt/min	Formula	Compounds	CAS	RI^1^	RI^2^	Source
1	4.75	C_5_H_10_O	Valeraldehyde	110-62-3	979	979	Ⅱ
2	6.15	C_7_H_8_	Toluene	108-88-3	1040	1042	Ⅰ Ⅱ
3	7.37	C_6_H_12_O	Hexanal	66-25-1	1082	1083	All
4	7.65	C_11_H_24_	n-Hendecane	1120-21-4	1091	1100	All
5	8.77	C_5_H_8_O	Trans-2-Pentenal	1576-87-0	1129	1127	Ⅰ Ⅱ
6	8.94	C_8_H_10_	P-xylene	106-42-3	1127	1138	All
7	10.09	C_5_H_10_O	1-Penten-3-ol	616-25-1	1169	1159	Ⅰ Ⅱ
8	10.17	C_10_H_16_	α-Terpinene	99-86-5	1171	1180	Ⅰ Ⅳ
9	10.83	C_10_H_16_	(+)-Dipentene	5989-27-5	1190	1195	Ⅰ Ⅱ Ⅳ
10	10.97	C_12_H_26_	Dodecane	112-40-3	1193	1200	All
11	11.77	C_6_H_10_O	Hex-2-enal	505-57-7	1217	1213	All
12	11.84	C_6_H_10_O	Trans-2-hexenal	6728-26-3	1218	1216	All
13	12.14	C_9_H_14_O	2-Pentylfuran	3777-69-3	1230	1231	All
14	12.51	C_10_H_16_	γ-Terpinene	99-85-4	1241	1246	Ⅰ Ⅱ Ⅲ
15	12.82	C_10_H_16_	Ocimene	13877-91-3	1250	1250	Ⅰ
16	13.02	C_9_H_12_	2-Ethyl toluene	611-14-3	1257	1258	Ⅰ
17	13.35	C_10_H_14_	P-Cymene	99-87-6	1271	1272	All
18	13.37	C_10_H_14_	M-Cymene	535-77-3	1266	1266	Ⅰ
19	13.52	C_10_H_14_	O-Cymene	527-84-4	1271	1275	Ⅱ
20	13.69	C_9_H_12_	1, 2, 4-Trimethylbenzene	95-63-6	1276	1283	All
21	13.79	C_10_H_16_	Cyclohexene	586-63-0	1279	-	Ⅰ
22	14.54	C_13_H_28_	n-Tridecane	629-50-5	1297	1300	All
23	15.24	C_7_H_12_O	Trans-2-Heptenal	18829-55-5	1321	1323	Ⅱ Ⅲ Ⅳ
24	15.24	C_7_H_12_O	2-Heptenal	57266-86-1	1321	1322	Ⅰ
25	15.39	C_5_H_10_O	Cis-2-penten-1-ol	1576-95-0	1326	1318	Ⅰ Ⅱ
26	15.52	C_9_H_12_	1, 2, 3-Trimethylbenzene	526-73-8	1330	1340	Ⅲ Ⅳ
27	15.66	C_8_H_14_O	6-Methyl-5-hepten-2-one	110-93-0	1335	1338	Ⅰ
28	16.50	C_6_H_14_O	1-Hexanol	111-27-3	1360	1355	Ⅰ Ⅱ
29	17.48	C_6_H_12_O	Leaf alcohol	928-96-1	1388	1382	Ⅰ Ⅱ
30	17.59	C_9_H_18_O	1-Nonanal	124-19-6	1392	1391	All
31	18.26	C_8_H_12_O	5-Ethyl-1-cyclopentene-1-carboxaldehyde	36431-60-4	1412	1410	All
32	18.75	C_8_H_14_O	(E)-2-Octenal	2548-87-0	1428	1429	All
33	18.93	C_10_H_12_	m, α-Dimethylstyrene	1124-20-5	1434	1440	Ⅰ Ⅱ Ⅲ
34	18.99	C_10_H_12_	α, P-Dimethylstyrene	1195-32-0	1437	1444	Ⅰ
35	19.63	C_8_H_16_O	1-Octen-3-ol	3391-86-4	1456	1450	All
36	19.70	C_2_H_4_O_2_	Acetic acid	64-19-7	1458	1449	Ⅰ
37	20.63	C_15_H_24_	(−)-α-Copaene	3856-25-5	1485	1492	Ⅰ Ⅱ
38	20.67	C_7_H_10_O	Trans, trans-2, 4-Heptadienal	4313-03-5	1487	1495	Ⅰ
39	20.96	C_10_H_20_O	Decyl aldehyde	112-31-2	1495	1498	Ⅱ
40	21.05	C_15_H_32_	n-Pentadecane	629-62-9	1498	1500	Ⅱ
41	21.56	C_7_H_6_O	Benzaldehyde	100-52-7	1515	1520	Ⅲ Ⅳ
42	22.05	C_9_H_16_O	Trans-2-nonenal	18829-56-6	1531	1534	Ⅰ Ⅳ
43	22.61	C_10_H_18_O	Linalool	78-70-6	1549	1547	Ⅰ
44	23.81	C_15_H_24_	β-Caryophyllene	87-44-5	1586	1595	Ⅱ Ⅲ
45	23.82	C_15_H_24_	(−)-Isocaryophyllene	118-65-0	1586	1587	Ⅰ
46	24.14	C_10_H_18_O	(−)-Terpinen-4-ol	20126-76-5	1596	1593	Ⅲ Ⅳ
47	24.16	C_10_H_18_O	Terpinen-4-ol	562-74-3	1596	1602	Ⅰ Ⅱ
48	24.56	C_10_H_16_O	β-Cyclocitral	432-25-7	1610	1611	All
49	24.65	C_8_H_16_O	Cis-5-octen-1-ol	64275-73-6	1613	1615	Ⅰ
50	25.12	C_8_H_8_O	Phenylacetaldehyde	122-78-1	1629	1640	All
51	25.29	C_10_H_18_O	3-Heptylacrolein	3913-81-3	1636	1644	Ⅱ
52	25.93	C_15_H_24_	α-Caryophyllene	6753-98-6	1657	1667	Ⅰ Ⅱ
53	26.89	C_9_H_14_O	Trans-2,4-Nonadienal	5910-87-2	1689	1700	Ⅰ Ⅱ Ⅳ
54	28.17	C_15_H_24_	α-Farnesene	502-61-4	1740	1746	Ⅰ
55	28.45	C_15_H_24_	(+)-δ-Cadinene	483-76-1	1745	1758	Ⅰ Ⅱ Ⅲ
56	28.62	C_10_H_16_O	(2E, 4Z)-Decadienal	25152-83-4	1759	1754	Ⅳ
57	28.76	C_10_H_22_O	Decyl alcohol	112-30-1	1764	1760	Ⅱ
58	30.71	C_6_H_12_O^2^	Hexanoic acid	142-62-1	1860	1846	Ⅳ
59	31.53	C^8^H^10^O	Phenethyl alcohol	60-12-8	1909	1906	Ⅱ
60	31.94	C^13^H^20^O	Irisone	14901-07-6	-	1971	Ⅲ
61	31.95	C^13^H^20^O	β-Lonone	79-77-6	1939	1940	Ⅰ Ⅱ
62	32.02	C_11_H_16_O	Jasmone	488-10-8	1944	1961	Ⅰ
63	34.83	C_10_H_12_O_2_	Eugenol	97-53-0	-	2169	Ⅰ Ⅱ
64	35.34	C_17_H_34_O_2_	Methyl palmitate	112-39-0	-	2208	Ⅰ Ⅱ

RI^1^: the standard retention index; RI^2^: the actual retention index.

**Table 3 foods-12-03335-t003:** Information about the qualitative VOCs detected by HS-GC-IMS.

No.	Rt/min	Formula	Compounds	CAS	RI^1^
1	1.49	C_3_H_8_O	Isopropyl alcohol	C67630	496.2
2	1.78	C_4_H_8_O	2-Methylpropanal monomer	C78842	544.3
3	1.79	C_4_H_8_O	2-Methylpropanal dimer	C78842	546.7
4	1.94	C_4_H_8_O_2_	Ethyl acetate monomer	C141786	572.4
5	2.07	C_4_H_8_O_2_	Ethyl acetate dimer	C141786	594.0
6	2.30	C_5_H_10_O	3-Methylbutanal monomer	C590863	633.8
7	2.34	C_5_H_10_O	3-Methylbutanal dimer	C590863	639.5
8	3.30	C_5_H_8_O	(E)-2-Pentenal dimer	C1576870	746.0
9	3.31	C_5_H_8_O	(E)-2-Pentenal monomer	C1576870	746.2
10	3.92	C_6_H_12_O	Hexanal dimer	C66251	793.0
11	4.48	C_6_H_12_O	Hexanal monomer	C66251	823.2
12	5.04	C_6_H_12_O	(E)-2-Hexen-1-ol dimer	C928950	853.7
13	5.26	C_6_H_12_O	(E)-2-Hexen-1-ol monomer	C928950	865.5
14	5.71	C_7_H_14_O	2-Heptanone dimer	C110430	889.8
15	5.74	C_7_H_14_O	2-Heptanone monomer	C110430	891.2
16	5.86	C_6_H_10_O	Cyclohexanone monomer	C108941	895.4
17	5.87	C_6_H_10_O	Cyclohexanone dimer	C108941	896.0
18	5.94	C_7_H_16_O	3-Heptanol	C589822	898.1
19	5.97	C_7_H_14_O	Heptanal dimer	C111717	899.3
20	6.04	C_7_H_14_O	Heptanal monomer	C111717	901.3
21	6.28	C_6_H_6_O_2_	2-Acetylfuran	C1192627	909.4
22	6.82	C_6_H_8_N_2_	Ethyl pyrazine	C13925003	927.1
23	7.14	C_10_H_16_	α-Pinene	C80568	937.8
24	7.61	C_7_H_12_O	(E)-Hept-2-enal monomer	C18829555	953.2
25	7.64	C_7_H_12_O	(E)-Hept-2-enal dimer	C18829555	954.0
26	7.65	C_7_H_6_O	Benzaldehyde monomer	C100527	954.2
27	7.66	C_7_H_6_O	Benzaldehyde dimer	C100527	954.7
28	8.35	C_8_H_14_O	1-Octen-3-one monomer	C4312996	977.3
29	8.36	C_8_H_14_O	1-Octen-3-one dimer	C4312996	977.6
30	8.48	C_8_H_16_O	Oct-1-en-3-ol	C3391864	981.8
31	8.78	C_9_H_14_O	2-Pentyl furan	C3777693	991.5
32	8.78	C_8_H_14_O	6-Methyl-hept-5-en-2-one	C110930	991.5
33	9.06	C_7_H_10_O	2,4-Heptadienal	C5910850	999.1
34	9.08	C_10_H_16_	α-Phellandrene	C99832	999.5
35	9.70	C_7_H_10_O	(E, E)-2,4-Heptadienal dimer	C4313035	1011.5
36	9.79	C_7_H_10_O	(E, E)-2,4-Heptadienal monomer	C4313035	1013.3
37	10.41	C_10_H_18_O	1.8-Cineole	C470826	1025.4
38	10.44	C_8_H_8_O	Phenylacetaldehyde monomer	C122781	1026.0
39	10.94	C_8_H_8_O	Phenylacetaldehyde dimer	C122781	1035.8
40	11.19	C_6_H_6_N_2_O	Acetyl pyrazine	C22047252	1040.8
41	11.55	C_10_H_16_	β-Ocimene	C13877913	1047.7
42	11.97	C_8_H_14_O	(E)-2-Octenal dimer	C2548870	1055.9
43	12.36	C_8_H_14_O	(E)-2-Octenal monomer	C2548870	1063.5
44	13.15	C_8_H_12_N_2_	2-Ethyl-3,5-dimethyl pyrazine	C13925070	1079.0
45	13.61	C_8_H_8_O_2_	Methyl benzoate	C93583	1088.0
46	14.27	C_10_H_18_O	Linalool	C78706	1101.0
47	14.57	C_9_H_18_O	Nonanal dimer	C124196	1106.7
48	14.72	C_9_H_18_O	Nonanal monomer	C124196	1109.8
49	17.06	C_10_H_18_O	Borneol monomer	C507700	1155.4
50	18.20	C_9_H_10_O_2_	Benzyl acetate monomer	C140114	1177.8

RI^1^: the standard retention index.

## Data Availability

The data presented in this study are available in the Appendix A.

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
