# Peer review of "An Exploration of Dynamic Changes in the Mulberry Growth Process Based on UPLC-Q-Orbitrap-MS, HS-SPME-GC-MS, and HS-GC-IMS"

_foods, 2023, doi:10.3390/foods12183335_

Round 1
Reviewer 1 Report
Shufang Wu et al. reported that non-volatile (n-VOCs) and volatile (VOCs) organic compound alterations during the ripening of mulberry. Various compound alterations were observed during ripening at different times. The experiments were well designed, and the results were interestingly presented. I only have a few comments, as followings;
1. The detailed method for metabolite identification should be presented in the method section. Are authentic standards and internal standards been used for structural confirmation and quantification?
2. The authors are suggested to discuss the role of important compounds alteration during ripening.
3. Information about the number of replicates used for compounds analyses should be included, as well as how were relativized/normalized the data for the visualization of percentage of component content.
4. Not only for food quality control, the authors are but also suggested to highlight the application in food safety/food authenticity as well as biological significance of those discoveries for example biochemical pathway or bioactive compound synthesis.
Reviewer 2 Report
The paper is interesting and discuss a very important issue, how to verify the quality of the foods. Mainly in vegetal materials is very important to can determine the markers that can give the main information about the quality of product - it was in right time harvested, it was properly processed, etc.
The design of study is very interesting and give us a lot of information well presented by the authors.
I have a difficulty to understood the antioxidant method used for evaluation. I do not understood what are A1 and A2, as absorbance. When time of analysis or for which kind of sample were registered? I suggest to authors to clarify this.
Reviewer 3 Report
The authors have presented a comprehensive and multi-directional analysis of the phytochemical profile found in fruits of the genus Morus, encompassing both volatile and non-volatile compounds. The analytical and statistical techniques utilized in this study were thoughtfully chosen, leading to reliable and meaningful results. I concur that the methodology, results description, discussion, and conclusions drawn by the authors are accurate and well-founded.
However, in order to further enhance the credibility and applicability of the obtained results, the authors should consider providing more comprehensive information about the plant material used in the study. Specifically, details such as the year of collection, prevailing environmental conditions during collection, and whether individual species were identified must be included. The taxonomy of the genus Morus is complex, thus precise identification of the analyzed taxa would undoubtedly augment the manuscript's value in practical applications.
Moreover, to strengthen the work, the authors could incorporate a section detailing the limitations of their study. Acknowledging potential constraints and uncertainties can provide readers with a clearer understanding of the scope and reliability of the findings. Additionally, the authors could discuss potential future research directions in this area, thereby inspiring further scientific investigations and expanding the field's knowledge base.
Nonetheless, the data presented in this study holds significant value, particularly with regards to the occurrence of natural compounds in Morus fruits and their correlation with the season of harvest. These findings possess wide-ranging applications across various industries, including but not limited to food and pharmaceuticals. The multidirectional applicability of this data underscores its relevance and importance in scientific and practical contexts.
The authors' efforts in presenting a meticulous analysis of Morus fruits' phytochemical profile using advanced analytical and statistical methods deserve commendation. However, further improvements could be made by providing detailed plant material identification, addressing study limitations, and offering insights into potential future research directions. Overall, this work contributes substantially to understanding of natural compounds in Morus fruits and their potential applications in diverse industries.
